# Knowledge of breastfeeding practice and associated factors among fathers whose wife delivered in last one year in Gurage Zone, Ethiopia

**Solomon Shitu**[1]*, **Daniel Adane**[1], **Haimanot Abebe**[2], **Ayenew Mose**[1], **Alex Yeshaneh**[1], **Bekele Beyene**[1], **Haile Workye**[3]

1 Department of Midwifery, College of Health and Medical Sciences, Wolkite University, Wolkite, Ethiopia,
2 Department of Public Health, College of Health and Medical Sciences, Wolkite University, Wolkite, Ethiopia,
3 Department of Nursing, College of Health and Medical Sciences, Wolkite University, Wolkite, Ethiopia

* solomonsht7@gmail.com

**Data Availability Statement:** All relevant data are within the manuscript and its Supporting information files.

## Abstract

### Background

Breastfeeding is the feeding of an infant or young child with breast milk directly from female human breasts. It confers short-term and long-term benefits for both child and mother, including helping to protect children against a variety of acute and chronic disorders. In mothers, breastfeeding (BF) reduces postpartum bleeding, enhances accelerated involution of the uterus, and plays a crucial role in child spacing. Fathers have an important but often neglected role in the promotion of healthy breastfeeding practices. Evidence shows that mothers who have a supportive and encouraging partner are more likely to plan to breast-feed for a longer duration. So, this study was aimed to assess knowledge and associated factors towards breastfeeding practice among fathers.

### Methods

A community-based cross-sectional study was conducted in Gurage Zone among 597 fathers. One stage cluster sampling technique was used to select study participants. An interviewer-administered questionnaire was used to collect the data and it was checked for consistency and completeness and entered into epi data and exported to SPSS for analysis. Bivariate and multivariate logistic regression analysis was done to identify independent predictors. P-value < 0.05 was considered to declare a result as statistically significant.

### Result

In this study, a total of 585 participants were involved making a response rate of 98%. The overall knowledge status of participants was 341 (58.3%). The mean age of participants was 29.5 (SD±4.5). Urban residence, educational status, exposure to media, having more than one baby at home, and accompany his wife during health-seeking were independent predictors of knowledge status.

**Funding:** The author(s) received no specific funding for this work.

**Competing interests:** The authors have declared that no competing interests exist.

**Abbreviations:** BF, Breastfeeding; CSA, Central Statistical Authority; EBF, Exclusive Breastfeeding; EDHS, Ethiopia Demographic and Health survey; FMOH, Federal Ministry of Health; HSDP, Health Sector Development Program; NNP, National Nutrition Program; SDM, Sustainable developmental goals; SNNPR, Southern Nation Nationality and Peoples Region; SPSS, Statistical Package for Social Science; WHO, World Health Organization.

## Conclusion

This study has shown the level of knowledge of fathers towards breastfeeding in the study area was low (58.3%). Residence, two or more babies at home, accompany during ANC, and indexed infant illness was independent predictors of knowledge status of fathers towards breastfeeding. Policymakers and possible stack holders should better focus on the improvement of knowledge because the knowledge determines the overall condition of the family including the psychological development of the children that affect their life especially in a country like Ethiopia in which most of the decisions are made by them. Other researchers focus on the interaction of parents and the child and feeding disorders.

## Introduction

Breastfeeding is the feeding of an infant or young child with breast milk directly from a female human breast. According to the world health organization (WHO), breast milk has the complete nutritional requirements that a baby needs for healthy development [1]. Exclusive breastfeeding (EBF) is to feed the infant only breast milk. Complementary feeding should be initiated after six months and continued for at least 24 months [1,2].

Breast milk is not only a good and safe nutritive source for the developing infant but also gives a rich source of immunity such as antibodies, leukocytes, growth factors, cytokines, and antimicrobial substances which support the immature immune system of the newborn till immunological maturity is attained [3]. Breastfeeding also confers short-term and long-term benefits for the mother it reduces postpartum bleeding, enhances accelerated involution of the uterus, and plays a crucial role in child spacing [4–6].

Worldwide Suboptimal breastfeeding contributes to 45% of neonatal infectious deaths, 30% of Diarrheal deaths, and 18% of acute respiratory deaths among under five years of age children in developing countries [7,8].

Non-breastfeeding results in an estimated 40% of under-five stunting in Western and Central Africa (WCA) and more than 60% in some other countries. In Ethiopia, suboptimal breastfeeding contributes to an estimate of 80,000 infant deaths per year [9–12]. Currently, the practice of exclusive breastfeeding in Ethiopia is low, and according to EDHS, 2016 report, only 58% of mothers with infants less than 6 months old breastfeed them exclusively. Among Ethiopia regions, the level of early initiation of breastfeeding and median duration of exclusive breastfeeding is minimal in Afar and Somalia region [13].

The study done by Cerniglia and colleagues (2014) showed that paternal involvement during breastfeeding predicts a good quality of father–infant interactions and the children score higher on Social Orientation. This interaction was originated from a good family psychological profile and the degree of involvement [14]. A recent study by Erris and colleagues (2020) showed that in the family's life cycle the relationships between parents and children have a significant influence on various aspects of everyday life [15]. Husband support can influence initiation, continuation, and exclusivity of feeding. And important in managing difficulties of feeding, and helping with household and child care duties [16].

A father's knowledge of breastfeeding is an important factor that contributes to the success of breastfeeding [17–19]. In Ethiopia, pregnancy and childbirth are often regarded as exclusively women's affairs. BF is the key area to improve child survival and promote healthy growth and development. Therefore, to reduce the incidence of infant mortality and morbidity

through increasing the husband's knowledge about BF is mandatory. As the knowledge of the authors, there is no evidence showing the knowledge status in the study area. So, this study was aimed to minimize the dearth of information in the area.

## Methods

### Study design, area, and period

A community-based cross-sectional study was conducted from December 10 to January 10 in the Gurage zone. The Gurage zone is one of the administrative zones of the south nation's nationalities and peoples of Ethiopia (SNNPR). It has 16 districts and 5 town administrations. Wolkite town is the capital city of the Gurage zone which is found 158KM far from Addis Ababa capital city of Ethiopia. According to the 2007 national household census, the Gurage zone has a total population of 1,279,646, of which 657,568 are women and 622078 are men.

### Population

All fathers whose wives gave birth in the last 1 year were source populations.

All fathers whose wife gave birth in the last 1year and living in selected districts and town administrations were the study population.

### Eligibility criteria

All fathers whose wives gave birth in the last 1 year in the study area and residents for at least six months in the zone were included.

Those fathers who were seriously ill and unable to respond during the data collection period were excluded.

### Sample size determination

The sample size for the study was calculated by using the formula for single population proportion by assuming 5% marginal error and 95% confidence interval ($\partial$ (alpha) = 0.05) and the prevalence of knowledge towards breastfeeding in Mish woreda which was 62% [20]. The calculated sample size was 362. By considering a design effect of 1.5 because the sampling procedure is population-based one-stage cluster sampling and a non-response rate of 10% were added to the calculated sample size. So the final sample size has come up to 597.

### Sampling procedure

A one-stage cluster sampling technique was used to select the participants. From a total of 16 districts and 5 town administrations, five districts and three town administrations were selected randomly by lottery method. From selected clusters, husbands with their wives delivered in the last one year were enrolled by systematic random sampling technique as study participants of this study.

### Data collection procedure and quality control

The data were collected by 15-degree holder data collectors. To ensure quality data collectors were trained about data collection techniques, procedures, and objectives for two days. The questionnaire was designed first in the English language and it was translated to local language Amharic language by a translator and again it was translated back to English. The questionnaire was adopted from different studies [9,10,12,16,19]. It contains socio-demographic data, questions assessing knowledge, some determinant factors of the father's knowledge towards

BF. The questions to assess knowledge were adopted from the previous study [20] which comprises 12 yes or no questions and the score given from 0 (minimum) to 12(maximum). Those who scored mean or above were leveled as knowledgeable and below the mean were not knowledgeable. A pretest was conducted on 5% of the total sample size at one health center which is not selected as a study area by data collectors and then the questionnaire was assessed for its clarity and a necessary correction was done accordingly. A structured interviewer-administered questionnaire was used to collect data.

Before interviewing data collectors gave information about the aim of the study, purposes, possible risks, and benefits, the right and refusal of mothers, and the confidentiality issues. During data, collection data collectors were supervised by supervisors, and overall activities were controlled by principal investigators, and finally after data collection before entry all collected data were checked for completeness.

## Dependent variables

Father's knowledge about breastfeeding.

## Independent variables

Socio-demographic factor (age, residence, income, educational status of both wife and husband, occupation of husband and wife, age at marriage).

Wife's obstetric variables (the previous history of infant illness, number of children (birth order), accompanied by his wife during MCH service, Source of information about BF.

## Operational definition

Breastfeeding is the feeding of an infant or young child with breast milk directly from female human breasts [1,21].

**Good knowledge.** Participants who scored mean and above considered having good knowledge [20,22–24].

**Poor knowledge.** Participants who scored mean and below were considered to have poor knowledge [20,22–24].

## Data processing and analysis

Data were coded, cleaned, edited, and entered into epi data version 3.1 to minimize logical errors, and then it was exported to SPSS software version 26 for further analysis. Descriptive and summary statistics were carried out. The result of the study was presented by using, summary measures such as percentages, mean and standard deviation. Bi-variate and multivariate logistic regression was carried out to identify the predictors associated with knowledge towards breastfeeding. All variables with a p-value of $\leq 0.25$ in bi-variant logistic regression were included in the multivariable model. Variables having a p-value $\leq$ of 0.05 in the multivariate analysis were taken as significant predictors of the outcome.

## Ethical consideration and consent to participate

Ethical clearance was obtained from Wolkite University, College of Health and Medical Sciences, Institutional Health Research Ethics Review Committee (IHRERC). A formal letter for permission and support was written to the zonal health department of Gurage from Wolkite University and official permission to undertake the study with the reference number of wku 00125/2020 was obtained and permission to conduct the study was asked. The respondents

were informed about the objective, purpose, risks, and benefits of the study and the right to refuse to participate, and then informed written and signed consent was taken.

The study posed a low or no more than minimal risk to the study participants. Also, the study did not involve any invasive procedures. Moreover, the confidentiality of information was guaranteed by using code numbers rather than personal identifiers and by keeping the data locked.

## Result

### Socio-demographic characteristics of the respondent

In this study, a total of 585 participants were involved making a response rate of 98%. The mean age of participants was 29.5 (SD±4.5) and the majority of participants were between the age group of 25–29. The majority of the respondents 344 (58.9%) were rural residents while the rest resides in urban areas. Most of the respondents, 520(88.9%), were married while 11 (1.9%) were co-habitants. Nearly half 272 respondents were orthodox religion followers. Thirty percent (179) of the respondent's wives ' occupations were merchants (Table 1).

### Obstetric characteristics of wife and family size

About two-thirds 383 (65%) of the participants were multiparous and 505 (84%) were two or more alive babies. About 456 (78.3%) of participants were ANC follow up at least once. From this 186 (40.7%) accompany their wives at least once. From participant's wives one-fourth 146 gave birth at home (Fig 1) and nearly two-third 373 (63.7%) of their wives delivered by SVD. Thirty-one percent 182 of neonates were ill and from those more than two-third of 131 (70.8%) were treated by in-patient or outpatient. From the study participants, 111 (18.6%) were PNC follow up from those more than three forth 94 (85%) of participants were not accompany during the visit (Table 2).

### Respondent's media exposure and source of information

Almost all of the respondents 566 (96.75%) have access to media in different ways. Of all participants, 38% got information about bf from health extension workers followed by 23.3% from health professionals and 15.8% from media (Fig 2).

### Substance abuse and social support

From the total study participants, nearly three fourth 429 (73.4%) were not substance abused while 540 (92.3%) of respondent's wives were not abused (Fig 3).

### Knowledge of respondents regarding breastfeeding

The overall prevalence of knowledge was 58.3% with 95%CI 53.4–62.6. The majority 480 (82%) of the respondents reported that breast milk is the first food to be given to infants after birth. About 370(63.3%) of respondents disagree that mothers should stop breastfeeding a sick infant. About more than half 344 (58.7%) were believed that exclusive breastfeeding may protect mothers from pregnancy in the first few months after birth.

### Predictors of husbands knowledge

Binary logistic regression was done to identify predictors for knowledge status. In the final model urban residence, having two or more babies, indexing infant illness, accompany ANC follow-up were independent predictors of the knowledge status of the participants. Urban

**Table 1. Sociodemographic characteristics of respondents for the study to assess knowledge of fathers whose wife delivered in the last 1year towards breastfeeding and its associated factors.**

| Variables | Category | Frequency | Percentage (%) |
|---|---|---|---|
| Age | 15–19 | 20 | 3.4 |
| | 20–24 | 156 | 26.6 |
| | 25–29 | 224 | 38.3 |
| | 30–34 | 88 | 15 |
| | 35–39 | 76 | 13 |
| | ≥40 | 21 | 3.6 |
| Marital status | Marries | 520 | 88.9 |
| | Unmarried | 7 | 1.2 |
| | Divorced | 47 | 8 |
| | Co habitant | 11 | 1.9 |
| Estimated monthly income | <500 | 66 | 11.3 |
| | 500–999 | 188 | 32.1 |
| | 1000–2000 | 201 | 36 |
| | >2000 | 130 | 22.2 |
| Occupation | Student | 21 | 3.6 |
| | Government employee | 178 | 30.4 |
| | Merchant | 164 | 28 |
| | Farmer | 182 | 31.1 |
| | Other | 10 | 1.7 |
| Occupation of the wife | Student | 34 | 5.8 |
| | Government employee | 164 | 28 |
| | Merchant | 179 | 30.6 |
| | Housewife | 132 | 22.5 |
| | Farmer | 66 | 11.3 |
| | Other | 10 | 1.7 |
| Religion | Orthodox | 272 | 46.5 |
| | Protestant | 93 | 15.9 |
| | Catholic | 71 | 12.1 |
| | Muslim | 143 | 24.4 |
| | Other | 6 | 1.02 |
| Educational level | Unable to read and write | 68 | 11.6 |
| | Read and write | 71 | 12.1 |
| | Primary | 144 | 24.6 |
| | Secondary | 109 | 18.6 |
| | >12 | 193 | 33 |
| Educational level wife | Unable to read and write | 81 | 13.8 |
| | Read and write | 101 | 17.3 |
| | Primary | 166 | 28.4 |
| | Secondary | 150 | 25.6 |
| | >12 | 87 | 14.9 |

resident participants were 4.23 knowledgeable than their counterparts (AOR = 4.23 95% CI: 2.33–7.22). Those husbands with two or more babies were 2.33 times more likely knowledgeable than those with one baby (AOR = 2.33 95%CI: 1.11–5.25). Those husbands who accompany during ANC and indexing baby illness were 3.31 and 5.32 times more knowledgeable

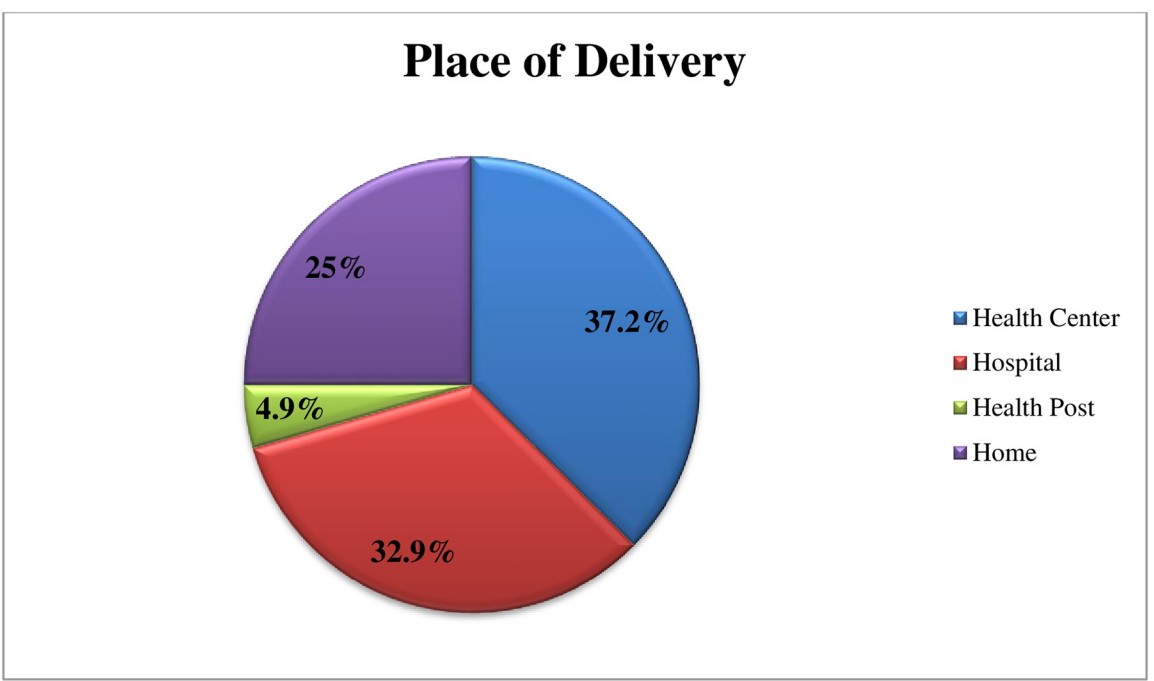

**Fig 1. Place of delivery of the wife for current baby in the study of to assess knowledge towards breastfeeding in the Gurage zone, southern Ethiopia.**

than their counterparts (AOR = 3.31 95% CI: 2.23–5.26) and (AOR = 5.25 95% CI: 3.20–8.34) respectively (Table 3).

## Discussion

In this study, the overall prevalence of knowledge status was 58.3% with 95%CI 53.8–61.1. This shows the knowledge status is low which needs better attention by different concerned

**Table 2. Obstetric characteristics of wife and family size of study participants to assess knowledge towards breastfeeding in Gurage Zone, Southern Ethiopia (n = 585).**

| Variables | Classification | Frequency | Percentage (%) |
|---|---|---|---|
| Parity the wife | Primiparus | 130 | 22.2 |
| | Multiparous | 383 | 65.5 |
| | Grand Multiparous | 72 | 12.3 |
| No of alive baby | 1 | 80 | 13.7 |
| | ≥2 | 505 | 86.3 |
| ANC follow up | Yes | 456 | 78 |
| | No | 129 | 22 |
| Accompany during ANC (456) | Yes | 186 | 40.8 |
| | No | 360 | 59.2 |
| Last infant illness | Yes | 182 | 31.1 |
| | No | 403 | 68.9 |
| PNC follow up of wife | Yes | 162 | 27.7 |
| | No | 423 | 73.3 |
| Accompany during PNC follow up (162) | Yes | 88 | 54.3 |
| | No | 74 | 45.7 |

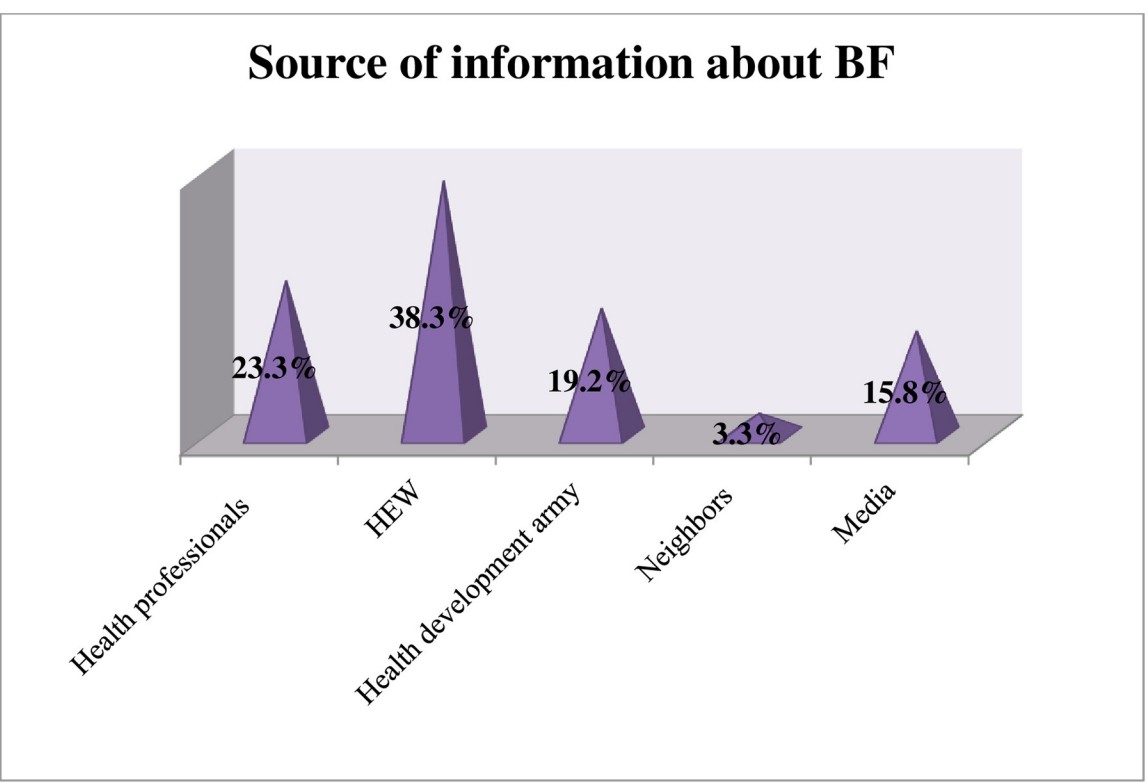

**Fig 2. Source of information of participants in the study to assess knowledge towards breastfeeding in the Gurage zone, southern Ethiopia.**

bodies like policymakers and health professionals. The finding was in line with the studies conducted in Kenya and northwest Ethiopia [19,20]. But the finding was lower than the study conducted in India and higher than the study conducted in afar Ethiopia [21,25]. This difference might be due to the difference in the study area, period, and access to health and media also is improved.

Those respondents living in urban areas were 4.23 times more likely knowledgeable than those who live in rural residents. This finding was in line with two findings done in Ethiopia [26,27]. The reason could be since living in urban is better access to information due to easy media access. And urbanization by itself helps them for better access to health care during pregnancy.

Participants who accompany his wife during ANC visit at least once were 3.31 times more likely knowledgeable than their counterparts. This result was in line with the study done in Kenya, northwest Ethiopia [19,20]. The reason might be due to the fact that those who go to health facilities during follow-up get health-related information from health professionals. And those who accompany his wife are most likely good knowledge and attitude for health-seeking, may also good educational background than their counterparts.

In this study fathers who had two or more babies in their home were 2.33 times more likely knowledgeable than those fathers who had only one baby. This finding was in line with studies done in Zambia and Nepal [28,29]. But the finding was incongruent with the finding in northwest Ethiopia [20]. This may be due to the reason when there is an increased number of a child there is increased knowledge due to increased exposure to different conditions. And experience by itself improves an individual's knowledge in different aspects.

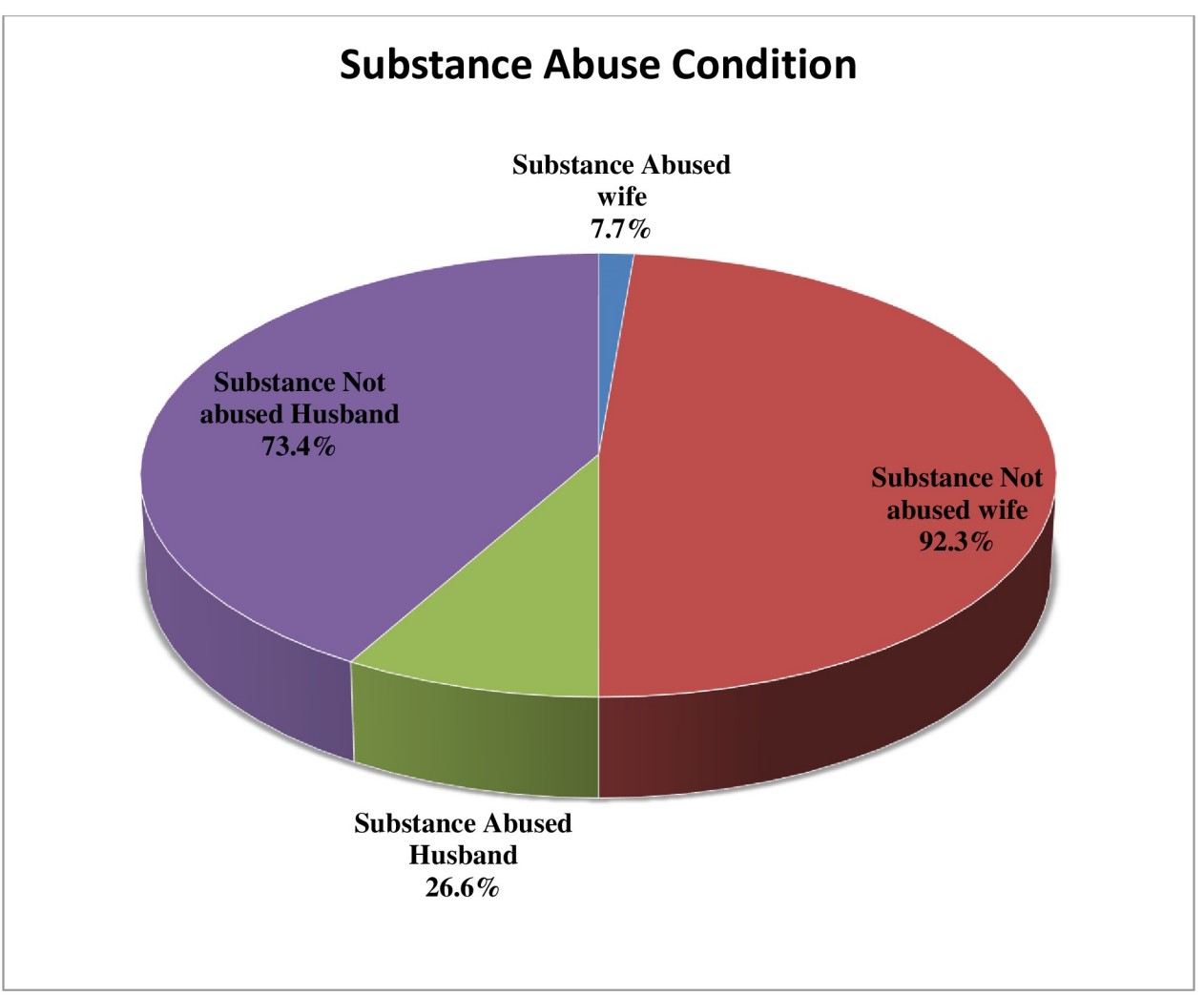

**Fig 3. History of substance abuse of study participants and their wives in the study to assess knowledge towards breastfeeding in the Gurage zone, Southern Ethiopia.**

A husband in whom their indexing baby develops illness was 5.32 times more likely knowledgeable than their counterparts. this study was supported by the cross-sectional study done in Pakistan and elsewhere in Africa on exploring father's role in BF practice shows those fathers with good family support are 3 times more likely to have a positive attitude toward BF as compared to their counterparts [30,31]. The reason might be a father with a sick baby may go to a health facility to seek care, at that time he may acquire information from health professionals, thus can improve knowledge.

## Conclusion

This study has shown the level of knowledge of fathers towards breastfeeding in the study area was low (58.3%). Residence, two or more babies at home, accompany during ANC, and indexed infant illness was independent predictors of knowledge status of fathers towards breastfeeding. Policymakers and possible stack holders should better focus on the

**Table 3. Predictors of knowledge of fathers whose wife delivered in the last 1year towards breastfeeding in Gurage Zone, Southern Ethiopia.**

| Variables | BF knowledge | | (95% CI) | |
|---|---|---|---|---|
| | Good knowledge | Poor knowledge | Crude OR | Adjusted OR |
| Residence | | | | |
| Urban | 163(47.8) | 78(32) | 1.95 (1.51–5.37) | 4.23 (2.33–7.22)* |
| Rural | 178(52.2) | 166(68) | 1 | 1 |
| Educational level | | | | |
| Unable to read & write68 | 20(5.8) | 48(19.6) | 0.86 (0.53–1.39) | 1.04 (0.59–1.80) |
| Read and write | 24(7) | 47(19.2) | 2.53 (0.45–4.66) | 2.68 (0.62–1.56) |
| Primary | 69(7.2) | 75(30.7) | 2.22 (0.37–2.66) | 3.24 (0.42–6.56) |
| Secondary | 84(20.2) | 25(10.2) | 2.64 (1.65–4.23) | 0.8 (0.22–2.90) |
| >12 | 144(42.2) | 49(19.7) | 1 | 1 |
| Parity of the wife | | | | |
| Primiparus130 | 76(22.3) | 54(22.1) | 2.91 (1.97–4.31) | 2.08 (0.30–3.66) |
| Multiparous383 | 225(66) | 158 (64.7) | 2.53 (1.37–4.66) | 2.68 (0.62–11.56) |
| Grand Multiparous72 | 40 (11.3) | 32 (13.2) | 1 | 1 |
| No of alive baby | | | | |
| ≥2 | 322 (94.4) | 183 (75) | 5.65 (1.44–9.51) | 2.33 (1.11–5.25)** |
| 1 | 19 (5.6) | 61 (25) | 1 | 1 |
| ANC follow up | | | | |
| Yes | 256 (75.1) | 200 (82) | 1.51 (1.12–4.87) | 1.09 (0.61–1.96) |
| No | 85(24.9) | 44 (18) | 1 | 1 |
| Accompany during ANC (456) | | | | |
| Yes | 124 (40.8) | 62 (25.6) | 2.00 (1.27–4.05) | 3.31 (2.23–5.26)*** |
| No | 180 (59.2) | 180 (74.6) | 1 | 1 |
| Last infant illness | | | | |
| Yes | 160 (46.9) | 22 (9) | 8.92 (5.81–12.58) | 5.25 (3.20–8.34)**** |
| No | 181 (53.1) | 222 (91) | 1 | 1 |
| Substance abuse | | | | |
| Yes | 41(12) | 115 (47.1) | 1 | 1 |
| No | 300 (88) | 129 (52.9) | 6.53 (3.34–7.25) | 2.02 (0.94–5.25) |

*Significant with P = 0.001,

** Significant with P = 0.000,

***Significant with P = 0.002 and

****Significant with P = 0.0021.

improvement of knowledge because the knowledge determines the overall condition of the family including the psychological development of the children that affect their life especially in a country like Ethiopia in which most of the decisions are made by them. Other researchers focus on the interaction of parents and the child and feeding disorders.

## Strength and limitation of the study

### Strength of study

As our knowledge is concerned this study is the first in this locality. The research was conducted at the community level that helped to access individuals who couldn't visit for different reasons.

### Limitation of study

Due to the cross-sectional study design, the study was exposed to a "chicken and egg" dilemma.

Better to check the practice of parental relationships with their children who may affect the nutritional, psychological, and whole development of the children.

## Supporting information

**S1 File. Questionnaire.**
(SAV)

**S2 File. Minimal data set.**
(DOCX)

## Acknowledgments

We would like to thank Wolkite University for giving technical support to prepare this research.

We would like to forward our deepest appreciation to in Gurage zone community administrations for their special support. Also, special gratitude goes to data collectors and our study participants.

## Declarations

The content of the study is solely the responsibility of the authors.

## Author Contributions

**Conceptualization:** Daniel Adane, Ayenew Mose, Haile Workye.

**Data curation:** Daniel Adane.

**Formal analysis:** Solomon Shitu.

**Investigation:** Solomon Shitu, Haimanot Abebe.

**Methodology:** Solomon Shitu, Haimanot Abebe, Ayenew Mose, Haile Workye.

**Software:** Alex Yeshaneh, Bekele Beyene.

**Writing – original draft:** Alex Yeshaneh, Bekele Beyene.

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
