## [Decision Letter · Decision Letter 0]

25 Jun 2021

PONE-D-21-03860

Knowledge of breast feeding practice and associated factors among fathers whose wife delivered in last one year in Gurage Zone, Ethiopia

PLOS ONE

Dear Authors,

Thank you for submitting your manuscript to PLOS ONE. After careful consideration, we feel that it has merit but does not fully meet PLOS ONE’s publication criteria as it currently stands. Therefore, we invite you to submit a revised version of the manuscript that addresses the points raised during the review process.

We look forward to receiving your revised manuscript.

Kind regards,

Luca Cerniglia, PhD

Academic Editor

PLOS ONE

Journal Requirements:

3. You indicated that you had ethical approval for your study. In your Methods section, please ensure you have also stated whether you obtained consent from parents or guardians of the minors included in the study or whether the research ethics committee or IRB specifically waived the need for their consent.

4. Please ensure that you refer to Figure 1 in your text as, if accepted, production will need this reference to link the reader to the figure.

Reviewers' comments:

Reviewer's Responses to Questions

**Comments to the Author**

1. Is the manuscript technically sound, and do the data support the conclusions?

Reviewer #1: Yes

Reviewer #2: Partly

2. Has the statistical analysis been performed appropriately and rigorously? 

Reviewer #1: Yes

Reviewer #2: Yes

3. Have the authors made all data underlying the findings in their manuscript fully available?

Reviewer #1: Yes

Reviewer #2: Yes

4. Is the manuscript presented in an intelligible fashion and written in standard English?

Reviewer #1: Yes

Reviewer #2: No

5. Review Comments to the Author

Reviewer #1: Dear editor,

Thank you very much for the invite to review the manuscript entitled: “Knowledge of breast feeding practice and associated factors among fathers whose wife delivered in last one year in Gurage Zone, Ethiopia”.

I have read with much interest the paper, since early experiences of nutrition are crucial for infants’ and young children’s physical and mental well-being.

I believe that the work of the authors can add as an interesting contribution to international empirical research focused on this topic.

The writing is adequately understandable and appears to be sound (form and contents are quite clear). The general aim and results are clearly recognized. The use of written English is good and clear.

My overall impression on all this manuscript is positive but I would like to comment on some points in the study, so that the authors can improve the final version of their work.

Introduction

The introduction should better emphasize the relevance of the topic in order to be linked to the main stated objective.

I suggest to better indicate the progression of the topics through the subdivision of the paper in a first part, dedicated to the presentation of theoretical aspects and empirical studies concerning the specific issue of breast feeding practice, highlighting the paternal parental figure, which is the key element of the whole work.

For this purpose the authors might consult the work by Cerniglia and colleagues (2014) and the work by Searle and colleagues (2020), listed below:

Cerniglia, L., Cimino, S., & Ballarotto, G. (2014). Mother–child and father–child interaction with their 24‐month‐old children during feeding, considering paternal involvement and the child's temperament in a community sample. Infant Mental Health Journal, 35(5), 473-481.

Searle, B. R. E., Harris, H. A., Thorpe, K., & Jansen, E. (2020). What children bring to the table: The association of temperament and child fussy eating with maternal and paternal mealtime structure. Appetite, 151, 104680.

Moreover, I think the introduction could be improved by indicating precisely the theoretical model that guides the authors’ methodological choices in their study, with particular reference to the developmental age. This important element should be stated in the work.

For this purpose the authors might consult the work by Erriu and colleagues (2020), in which, for istance, the theoretical and empirical model of Developmental Psychopathology is presented.

Erriu, M., Cimino, S., & Cerniglia, L. (2020). The Role of Family Relationships in Eating Disorders in Adolescents: A Narrative Review. Behavioral Sciences, 10(4), 71.

Method

The section on methodology could be improved in the choice of titles to be given to subsections.

A possible articulation could be the following:

Research Methods

-Subjects and procedure

-Measures

-Statistical analysis

With regard to the instruments and procedures, information on the properties of the instruments and a specific description of procedure should be added.

Discussion and conclusion

In the conclusions section authors should discuss more precisely the complexity of father’s role in relation to the well-being of the child.

In this attempt authors should consider some factors that could be take into account in future research extensions, such as the interplay between social, biological and psychological factors.

Reviewer #2: Thank you for the opportunity of revising this manuscript titled “Knowledge of breast feeding practice and associated factors among fathers whose wife delivered in last one year in Gurage Zone, Ethiopia”. I think that the paper focuses on interesting aspects, but there were some limitations. So I think that it can be published in this Journal, but with minor revision. Please find below some comments.

INTRODUCTION

It would be important for the authors to describe the specific characteristics by which breastfeeding can be defined as suboptimal.

The authors should report in more detail on previous studies that explored the father's role in breastfeeding of their children and on the quality of early father-children feeding interaction with a focus on possible risk and protective factors associated.

To this end, the authors should cite the following recent works:

- Ogbo, F. A., Akombi, B. J., Ahmed, K. Y., Rwabilimbo, A. G., Ogbo, A. O., Uwaibi, N. E., ... & Agho, K. E. (2020). Breastfeeding in the community—how can partners/fathers help? A systematic review. International journal of environmental research and public health, 17(2), 413.

- Cimino, S., Marzilli, E., Tafà, M., & Cerniglia, L. (2020). Emotional-Behavioral Regulation, Temperament and Parent–Child Interactions Are Associated with Dopamine Transporter Allelic Polymorphism in Early Childhood: A Pilot Study. International Journal of Environmental Research and Public Health, 17(22), 8564.

- Ng, R. W. L., Shorey, S., & He, H. G. (2019). Integrative review of the factors that influence fathers’ involvement in the breastfeeding of their infants. Journal of Obstetric, Gynecologic & Neonatal Nursing, 48(1), 16-26.

It is important to clarify what this study adds to the previous literature. Why is it important?

There is no clear theoretical framework and the authors should describe from the beginning the theoretical framework from which they start for their own study.

At the end of the introduction, the authors should clearly describe the objectives of the study and the underlying assumptions, based on previous literature

METHODS

Please provide additional details regarding participant consent and opinion of the ethics committee, including protocol number.

The authors need to provide more information on the instrument used to assess Father's knowledge about breastfeeding. Has an ad hoc self-report questionnaire been constructed? How many and which items? Was a clinical interview conducted? On the basis of which criteria is Father's knowledge about breastfeeding knowledge considered good or poor?

Among the independent variables the authors considered the following Socio-demographic factor: age, residence, income, educational status of both wife and husband, occupation of husband and wife, age at marriage.

Why? It is important that the authors mention in the introduction the previous literature that has suggested the role played by these variables.

DISCUSSION

Authors should more clearly discuss the findings from the previously identified literature and

present conclusions supported by the results.

6. PLOS authors have the option to publish the peer review history of their article (what does this mean?). If published, this will include your full peer review and any attached files.

Reviewer #1: No

Reviewer #2: No

---

## [Author Response · Author response to Decision Letter 0]

1 Jul 2021

Author’s Point-by-Point Response to the Reviewer's and Editors Reports

Title: Knowledge of breastfeeding practice and associated factors among fathers whose wife delivered in last one year in Gurage Zone, Ethiopia

Corresponding author: Solomon Shitu Ayen (solomonsht7@gmail.com)

Authors

Daniel Adane 

Haimanot Abebe 

Ayenew Mose 

Alex Yeshaneh

Bekele Beyene

Haile Workye 

ID: PONE-D-21-03860

Journal: PLOS ONE

 Article type: Research article

First of all, the authors would like to thank PLOS ONE Journal editors and the respective reviewers for reviewing our manuscript and providing the necessary comments to be corrected. As per the comments given, we have made corrections point by point to comment. The authors tried to answer all the issues raised by the editorial team and reviewers. Please note that we gave our response in blue font color.

Point by point response to editor

Comment “Please review your reference list to ensure that it is complete and correct. If you have cited papers that have been retracted, please include the rationale for doing so in the manuscript text, or remove these references and replace them with relevant current references. Any changes to the reference list should be mentioned in the rebuttal letter that accompanies your revised manuscript. If you need to cite a retracted article, indicate the article’s retracted status in the References list and also include a citation and full reference for the retraction notice

Response 1: Thank you for your great suggestion and timely comments. Corrected 

Comment 2: Please ensure that your manuscript meets PLOS ONE's style requirements, including those for file naming. The PLOS ONE style templates can be found at

Response 2: Thank you for your great suggestion and timely comments. Corrected

Comment 3: We suggest you thoroughly copyedit your manuscript for language usage, spelling, and grammar. If you do not know anyone who can help you do this, you may wish to consider employing a professional scientific editing service.

Response 3: Thank you for your great suggestion and timely comments. We have taken corrections concerning to any English errors and typos by using online grammar and English typo correctors apps (We used the following links- 

https://app.grammarly.com/?network=g&utm_source=google&matchtype=e&gclid=Cj0KCQjwo-aCBhC-ARIsAAkNQiuJ49UHhl6ibhQfzq9D4wGrbSOeZPv49UoRqSnd4ThQ-KKrPp uBp4aAjLgEALw_wcB&placement=&q=brand&utm_content=486649398671&gclsrc=aw.ds&utm_campaign=brand_f1&utm_medium=cpc&utm_term=grammarly and https://pubsure.researcher.life/author/?active_tab=recent_plan

So, now we have solved all iniquities related with the English language including the tense used and unnecessary capitalization and other typos/ errors 

Comment 4: You indicated that you had ethical approval for your study. In your Methods section, please ensure you have also stated whether you obtained consent from parents or guardians of the minors included in the study or whether the research ethics committee or IRB specifically waived the need for their consent.

Response 4: Thank you for your great suggestion and timely comments. Corrected 

Comment 5: Please ensure that you refer to Figure 1 in your text as, if accepted, production will need this reference to link the reader to the figure.

Response 5: Thank you for your great suggestion and timely comments. Type error corrected

Comment 5: Please include captions for your Supporting Information files at the end of your manuscript, and update any in-text citations to match accordingly. Please see our Supporting Information guidelines for more information: http://journals.plos.org/plosone/s/supporting-information.

Response 5: Thank you for your great suggestion and timely comments. Corrected 

Point by point response to Reviewer# 1

Dear, 

Comment 1: Introduction

The introduction should better emphasize the relevance of the topic in order to be linked to the main stated objective.

I suggest bettering indicating the progression of the topics through the subdivision of the paper in a first part, dedicated to the presentation of theoretical aspects and empirical studies concerning the specific issue of breast feeding practice, highlighting the paternal parental figure, which is the key element of the whole work.

For this purpose the authors might consult the work by Cerniglia and colleagues (2014) and the work by Searle and colleagues (2020), listed below

Response 1: Thank you for your great suggestion and timely comments. After reading suggested research papers some amendment done

Comment 2: Moreover, I think the introduction could be improved by indicating precisely the theoretical model that guides the authors’ methodological choices in their study, with particular reference to the developmental age. This important element should be stated in the work.

For this purpose the authors might consult the work by Erriu and colleagues (2020), in which, for istance, the theoretical and empirical model of Developmental Psychopathology is presented.

Response 2: Thank you for your great suggestion and timely comments. I tried to read the document which is related with adolescents and their psychological development. And I incorporated some concepts in the introduction but it is difficult to develop conceptual framework. 

Comment 3: Method

The section on methodology could be improved in the choice of titles to be given to subsections.

A possible articulation could be the following:

Research Methods

-Subjects and procedure

-Measures

-Statistical analysis

With regard to the instruments and procedures, information on the properties of the instruments and a specific description of procedure should be added.

Response 3: Thank you for your great suggestion and timely comments. Corrected according to your suggestion and journals requirement 

Comment 4: Discussion and conclusion

In the conclusions section authors should discuss more precisely the complexity of father’s role in relation to the well-being of the child.

In this attempt authors should consider some factors that could be take into account in future research extensions, such as the interplay between social, biological and psychological factors

Response 4: Thank you for your great suggestion and timely comments. Corrected accordingly. 

Point by point response to Reviewer# 2

Dear, 

Comment 1: INTRODUCTION

It would be important for the authors to describe the specific characteristics by which breastfeeding can be defined as suboptimal.

The authors should report in more detail on previous studies that explored the father's role in breastfeeding of their children and on the quality of early father-children feeding interaction with a focus on possible risk and protective factors associated.

Response 1: Thank you for your great suggestion and timely comments. Corrected as your suggestion

Comment 2: It is important to clarify what this study adds to the previous literature. Why is it important?

There is no clear theoretical framework and the authors should describe from the beginning the theoretical framework from which they start for their own study.

At the end of the introduction, the authors should clearly describe the objectives of the study and the underlying assumptions, based on previous literature

Response 2: Thank you for your great suggestion and timely comments. Corrected as, “Therefore, to reduce the incidence of infant mortality and morbidity through increasing the husband's knowledge about BF is mandatory. As the knowledge of the authors, there is no evidence showing the knowledge status in the study area. So, this study was aimed to minimize the dearth of information in the area”.

Comment 3: METHODS

Please provide additional details regarding participant consent and opinion of the ethics committee, including protocol number

Response 3: Thank you for your great suggestion and timely comments. Corrected 

“Ethical consideration and consent to participate

Ethical clearance was obtained from Wolkite University, College of Health and Medical Sciences, Institutional Health Research Ethics Review Committee (IHRERC). A formal letter for permission and support was written to the zonal health department of Gurage from Wolkite University and official permission to undertake the study with reference number of wku 00125/2020 was obtained and permission to conduct the study was asked. The respondents were informed about the objective, purpose, risks, and benefits of the study and the right to refuse to participate, and then informed written and signed consent was taken.

The study posed a low or no more than minimal risk to the study participants. Also, the study did not involve any invasive procedures. Moreover, the confidentiality of information was guaranteed by using code numbers rather than personal identifiers and by keeping the data locked.”

Comment 4: The authors need to provide more information on the instrument used to assess Father's knowledge about breastfeeding. Has an ad hoc self-report questionnaire been constructed? How many and which items? Was a clinical interview conducted? On the basis of which criteria is Father's knowledge about breastfeeding knowledge considered good or poor?

Response 4: Thank you for your great suggestion and timely comments. Corrected as, “The data were collected by 15-degree holder data collectors. To ensure quality data collectors were trained about data collection techniques, procedures, and objectives for two days. The questionnaire was designed first in the English language and it was translated to local language Amharic language by a translator and again it was translated back to English. The questionnaire was adopted from different studies [9, 10, 12,16, 18]. It contains socio-demographic data, questions assessing knowledge, some determinants factors of father's knowledge towards BF. The questions to assess knowledge were adopted from previous study [17] which comprises of 12 yes or no questions and the score given from 0 (minimum) to 12(maximum). Those who scored mean or above were leveled as good knowledge and below the mean were poor knowledge. A pretest was conducted on 5% of the total sample size at one health center which is not selected as a study area by data collectors and then the questionnaire was assessed for its clarity and a necessary correction was done accordingly. A structured interviewer administered questionnaire was used to collect data.

Before interviewing data collectors gave information about the aim of the study, purposes, possible risks, and benefits, the right and refusal of mothers, and the confidentiality issues. During data, collection data collectors were supervised by supervisors, and overall activities were controlled by principal investigators, and finally after data collection before entry all collected data were checked for completeness.”

Comment 5: Among the independent variables the authors considered the following Socio-demographic factor: age, residence, income, educational status of both wife and husband, occupation of husband and wife, age at marriage.

Why? It is important that the authors mention in the introduction the previous literature that has suggested the role played by these variables.

Response 5: Thank you for your great suggestion and timely comments. Of course these variables have mentioned as associated factors in the previous studies and some of the variables were added from context point of view.

Comment 6: DISCUSSION

Authors should more clearly discuss the findings from the previously identified literature and

present conclusions supported by the results

Response 6: Thank you for your great suggestion and timely comments. Rewritten as your suggestion.

---

## [Editor Report · Decision Letter 1]

5 Jul 2021

Knowledge of breast feeding practice and associated factors among fathers whose wife delivered in last one year in Gurage Zone, Ethiopia

PONE-D-21-03860R1

Dear Authors,

We’re pleased to inform you that your manuscript has been judged scientifically suitable for publication and will be formally accepted for publication once it meets all outstanding technical requirements.

Kind regards,

Luca Cerniglia, PhD

Academic Editor

PLOS ONE

Additional Editor Comments (optional):

The authors have successfully responded to the reviewers' comments.

I think the paper can be accepted for publication.

Best regards
---

## [Editor Report · Acceptance letter]

9 Jul 2021

PONE-D-21-03860R1 

Knowledge of breastfeeding practice and associated factors among fathers whose wife delivered in last one year in Gurage Zone, Ethiopia 

Dear Dr. Shitu:

I'm pleased to inform you that your manuscript has been deemed suitable for publication in PLOS ONE. Congratulations! Your manuscript is now with our production department. 

Kind regards, 

on behalf of

Dr. Luca Cerniglia 

Academic Editor

PLOS ONE